# Signal Structure-Aware Gaussian Splatting for Large-Scale Scene Reconstruction

**Weiyi Xue**[1,2]**, Fan Lu**[1†]**, Chi Zhang**[4]**, Tianhang Wang**[1]**,**
**Sanqing Qu**[1]**, Zehan Zheng**[5]**, Boyuan Zheng**[1]**, Junqiao Zhao**[1]**, Guang Chen**[1,2,3†*]
[1] Tongji University, [2] Shanghai Innovation Institute, [3] Shanghai Westwell Technology Co., Ltd
[4] Beijing University of Technology, [5] Johns Hopkins University
[†] Corresponding author

## Abstract

3D Gaussian Splatting has demonstrated remarkable potential in novel view synthesis. In contrast to small-scale scenes, large-scale scenes inevitably contain sparsely observed regions with excessively sparse initial points. In this case, supervising Gaussians initialized from low-frequency sparse points with high-frequency images often induces uncontrolled densification and redundant primitives, degrading both efficiency and quality. Intuitively, this issue can be mitigated with scheduling strategies, which can be categorized into two paradigms: modulating target signal frequency via densification and modulating sampling frequency via image resolution. However, previous scheduling strategies are primarily hardcoded, failing to perceive the convergence behavior of the scene frequency. To address this, we reframe scene reconstruction problem from the perspective of signal structure recovery, and propose SIG, a novel scheduler that **S**ynchronizes **I**mage supervision with **G**aussian frequencies. Specifically, we derive the average sampling frequency and bandwidth of 3D representations, and then regulate the training image resolution and the Gaussian densification process based on scene frequency convergence. Furthermore, we introduce Sphere-Constrained Gaussians, which leverage the spatial prior of initialized point clouds to control Gaussian optimization. Our framework enables frequency-consistent, geometry-aware, and floater-free training, achieving state-of-the-art performance with a substantial margin in both efficiency and rendering quality in large-scale scenes.

## 1 Introduction

High-fidelity and real-time Novel View Synthesis (NVS) in large scale scenes is a fundamental requirement for a wide range of applications, including UAV navigation, autonomous driving. Recently, Neural Radiance Fields (NeRFs) (Mildenhall et al., 2021) have made significant progress in NVS, yet suffer from prohibitive optimization and rendering costs. In contrast, 3DGS (Kerbl et al., 2023) achieves comparable fidelity with fast rendering capability by modeling scenes with Gaussian primitives. Nonetheless, even with block-wise parallelism (Liu et al., 2024), training remains inefficient in city-scale scenes. The massive inputs and numerous Gaussians, alongside the regions lacking initial point clouds and sparsely observed, lead to redundancy and floaters, thereby compromising both efficiency and reconstruction quality.

Delving deeper into the issue, given that large-scale scenes inevitably require more image supervision and Gaussians, the degradation can be further attributed to the imbalance between the *rendering resolution* and the *densification strategy*. This can be further elaborated from two complementary perspectives: (1) The common practice of rendering high-resolution images throughout training imposes substantial computation and memory overhead. (2) Supervising low-frequency-initialized Gaussians with high-frequency images leads to uncontrolled densification and redundant Gaussians, as excessive gradients induce premature growth focused on fine textures, while overlooking the underlying geometric structure.

---

[*]Project leader, supervised the work and defined the conceptualization

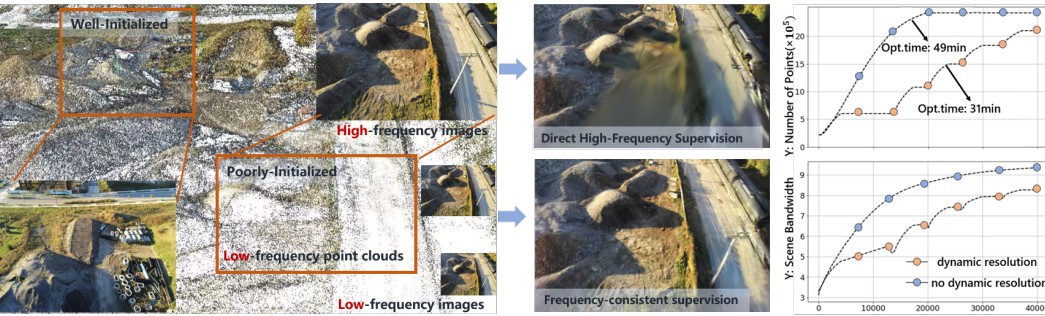

Figure 1: **Floaters and redundancy.** Prior methods that directly supervise with high-frequency images lead to redundancy and cannot exploit more primitives to capture high-frequency details.

We recognize that the *rendering resolution* and the *densification strategy* are not orthogonal concepts, and seek to address the challenges through a unified framework. By reframing scene reconstruction from the perspective of signal structure recovery, common schedulers for densification and resolution can be regarded as controlling the target signal's frequency content and the sampling frequency. However, such strategies raise a fundamental question: *when should the resolution be increased and densification be performed?* Some preliminary attempts have been made in prior works. DashGS (Chen et al., 2025) progressively increases image resolution in a non-linear fashion over training iterations, whereas methods such as TamingGS (Mallick et al., 2024), rely on predefined densification schedules. Since these schedulers are predetermined before training rather than being adapted during the optimization, we refer to such approaches as *hard-coded scheduling strategies*. However, the sampling frequency and the fidelity of signal reconstruction are inherently coupled. Hard-coded scheduling may impede effective learning by imposing premature high-resolution supervision or delaying essential refinements. Ideally, resolution should be increased only when further training at the current level ceases to yield significant improvements. This occurs when the spectral content encoded by the Gaussians reaches the maximum recoverable frequency (*i.e.*, the Nyquist frequency), thereby allowing higher-resolution images to guide the densification process in recovering high-frequency details.

Building on the above discussion, to ensure frequency-consistent optimization, it is essential to characterize the frequency of the sampling and target signals. Thus, we first mathematically derive the representation frequency of 3D Gaussians. This guides the supervision resolution according to the evolving scene frequency. By leveraging low resolution for structure recovery and high resolution for texture refinement, we enable adaptive resolution adjustment during training. Beyond the frequency inconsistency, unconstrained Gaussian movement and scaling in prior methods may lead to the neglect of structural priors. While Neural Gaussians (Lu et al., 2024) introduce neural anchors, they remain limited by block-partitioned training with shared MLP decoders, hindering scalability and stability in large scale scenes. To leverage the spatial priors exhibited by point clouds, we further introduce Sphere-Constrained Gaussians to reduce redundancy and preserve scene structure, where all Gaussians are confined within a sphere based on density priors.

In summary, our contributions are as follows: (1) We mathematically define the average frequency of 3D Gaussians representation and propose a novel scheduler that synchronizes image supervision with gaussian frequency, to mitigates redundancy and accelerates training. (2) We propose Sphere-Constrained Gaussians, which leverage structural priors to restrict the optimization space. (3) Our framework achieves substantial improvements in quality (+0.9 dB PSNR) and training speed (1.5× per block) across multiple benchmarks.

## 2 RELATED WORK

**Novel View Synthesis and 3D Representation.** NeRF (Mildenhall et al., 2021) and its related works (Gao et al., 2022; Chen et al., 2021; Barron et al., 2021) have achieved remarkable progress in NVS. To achieve faster training and rendering, NeRFs have gradually incorporated explicit structures (Chen et al., 2022; Müller et al., 2022), while remain computationally expensive due to the

dense sampling along each ray. In contrast, 3DGS (Kerbl et al., 2023) adopts a purely explicit representation, achieving photorealistic rendering quality while significantly improving rendering speed. Subsequent works have further enhanced 3DGS by incorporating techniques such as anti-aliasing (Yu et al., 2024), training acceleration, and block-wise optimization (Liu et al., 2024; Lin et al., 2024). More recent GS-based approaches introduce neural Gaussians (Lu et al., 2024), combining implicit representations to improve adaptability. However, the implicit components compromises rendering efficiency and hinders block-wise optimization in large-scale scenes.

**Scheduling Strategy for 3DGS Optimization.** Multiscale representations (Müller et al., 2022) are commonly employed in NeRF to enable coarse-to-fine optimization (Xue et al., 2024; Lin et al., 2021), which allows a progressive exposure of high-resolution features during training. In 3DGS, the training scheduler can regulate the optimization via Gaussian densification and image resolution. For the former, prior works such as TamingGS (Mallick et al., 2024; Rota Bulò et al., 2024) and DashGS (Chen et al., 2025) primarily focus on controlling Gaussian densification to limit the number of primitives in early training. For the latter, DashGS reduces computational cost by dynamically adjusting the rendering resolution throughout training. However, prior scheduling of either resolution or primitive growth is predefined and static throughout the optimization. This lack of adaptivity means that the training process cannot respond to the actual reconstruction progress. Premature introduction of high-resolution supervision may destabilize early optimization stages, while delayed scheduling may delay convergence and underutilize computational resources.

**Large-scale Scene Reconstruction.** Recent advances in large-scale scene reconstruction predominantly follow a divide-and-conquer paradigm. NeRF-based approaches such as Block-NeRF (Tancik et al., 2022; Zhenxing & Xu, 2022) and Mega-NeRF (Xu et al., 2024; Turki et al., 2022) partition the scene into spatial blocks, yet rendering latency remains a critical limitation. In contrast, 3DGS has emerged as a more efficient alternative, with many studies focusing on block-wise representations that incorporate level-of-detail rendering. VastGS (Lin et al., 2024) performs progressive partitioning based on camera distribution. DOGS (Chen & Lee, 2025) and BlockGS (Wu et al., 2025) adopt recursive strategies to ensure balanced computation across blocks. CityGS (Liu et al., 2024) leverages coarse Gaussians to guide scene partitioning and performs direct merging of Gaussians within blocks. Octree-GS (Ren et al., 2024) initializes the scene with structured neural anchors and decodes gaussian using MLP-decoder, while lacks native support for block-wise training due to the shared MLP. MomentumGS (Fan et al., 2024) attempts to address this issue but incurs frequent inter-block synchronization. Despite these advances, challenges such as Gaussian redundancy and floater artifacts persist in large-scale scenes, hindering both reconstruction quality and rendering efficiency.

## 3 METHOD

We first briefly revisit 3DGS in Section 3.1. Then, we formalize average sampling frequency and scene frequency bandwidth in Section 3.2, and propose our frequency-aligned resolution and densification scheuler in Section 3.3. Section 3.4 introduces our spatial-aware Gaussian optimization process.

### 3.1 PRELIMINARIES

3DGS utilizes Gaussian primitives $\{\mathcal{G}_i(\mathbf{x}) = \exp\left(-\frac{1}{2}(\mathbf{x} - \mathbf{p}_i)^\top \Sigma_i^{-1}(\mathbf{x} - \mathbf{p}_i)\right)\}_{i=1}^N$ to represent a 3D scene, where $\Sigma_i \in \mathbb{R}^{3 \times 3}$ denotes the covariance matrix, and $\mathbf{p}_i \in \mathbb{R}^3$ represents the position. Each primitive also possesses an opacity $o_i$ and color attributes $\mathbf{c}_i$. Per-pixel colors are obtained via $\alpha$-blending: $\mathbf{C} = \sum_{i=1}^N \alpha_i \mathbf{c}_i \prod_{j=1}^{i-1}(1 - \alpha_j)$, $\alpha$ denotes the transparency weight, which is derived from $o_i$ and $\Sigma_i$. We adopt the partitioning strategy of CityGS, using coarse training to build a scaffold and fine training for each block.

### 3.2 RETHINKING THE SCHEDULER FROM SIGNAL STRUCTURE RECOVERY

**Theoretical Foundations.** Reconstruction can be viewed as the task of recovering continuous 3D signals from discretely sampled images. Existing methods typically schedule the training process along two axes: (1) adjusting the frequency of the 3D Gaussians via controlled densification; (2)

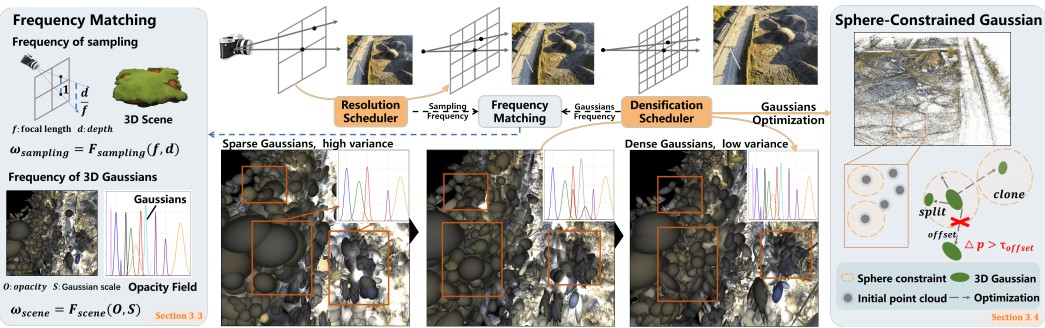

Figure 2: **Overview.** We define Gaussian frequency based on the opacity field, represented as a weighted sum of Gaussians. Using a frequency-matching module, we synchronize image supervision with Gaussian frequencies (SIG), and optimize with Sphere-Constrained Gaussians to incorporate geometric priors.

modulating the frequency of the supervision signal. For instance, DashGS computes the frequency of images sampled at different resolutions and linearly maps the frequency at different image scales to the training iterations. TamingGS focus on controlling densification. However, they all rely on hard-coded schedulers fixed prior to optimization. Consequently, premature introduction of high-resolution supervision can destabilize early optimization, while delayed scheduling may slow convergence and lead to inefficient use of computational resources.

According to the Nyquist Sampling Theorem, sampling at frequency $f$ limits recoverable components to $[0, f/2]$, known as the signal bandwidth. Given a sampling frequency $f_{img}$, the effective bandwidth of the scene will converge to a fixed value $B_{scene}$ (under certain assumptions in Appendix). Consequently, the sampling frequency should be increased once the current bandwidth converges, which indicates sufficient recovery. Once the relationship between the two is established, the key challenge is thus to determine the sampling frequency and the effective bandwidth of the 3D Gaussians.

**Sample Frequency.** For focal length $f$ and sampling depth $d$, a unit screen-space interval corresponds to a 3D space with radius of $d/f$. Assuming constant depth, the sampling frequency of the entire image is proportional to $f/d$. Downsampling an image by factor $t$ scales the focal length to $f' = f/t$, yielding a sampling frequency $v' = v/t$. Therefore, changing the image resolution modifies the sampling frequency of the signal. In practice, each image samples only a subset of the scene, and varying depth $d$ causes non-uniform sampling frequencies. From a differential viewpoint, each local patch can be approximated as uniformly sampled. We thus define the average sampling frequency over the scene as:

$$v = \sum_{i=1}^{n} \int_s w_i(s) \cdot \frac{f}{d_i(s)}\, ds, \quad \int_s w_i(s) ds = 1. \tag{1}$$

where $s$ denotes a local patch, $w_i(s)$ is the weight, representing the contribution of patch $s$ to whole scene, and $w_i(s) = 0$ if $s$ is not observed. $d_i(s)$ denotes the depth of patch $s$ in the $i$-th view. Clearly, this average frequency increases with image number $n$ and exhibits a *strict linear dependence on the focal length $f$*.

**Scene Signal Bandwidth.** Based on the above discussion, local patch frequencies are inherently non-uniform. Fortunately, our goal is to recover the dominant structures of the scene signal. In this context, the average signal bandwidth—computed over all local patches—serves as a global measure of the scene's frequency content, and naturally corresponds to the average sampling frequency in Eq. (1). Geometrically, variations in the scene's density function capture the spatial frequency of its structure. We explicitly write the scene opacity field as a weighted sum of 3D Gaussians:

$$D(\mathbf{x}) = \sum_{i=1}^{n} o_i\, G_i(\mathbf{x}) = \sum_{i=1}^{n} o_i\, (2\pi)^{3/2} \det(\mathbf{\Sigma})^{1/2} \mathcal{N}(\mathbf{x}; \mu_i, \Sigma_i), \tag{2}$$

where $G_i$ denotes the unnormalized Gaussian centered at $\mu_i$ with covariance $\Sigma_i$ and opacity $o_i$. $\mathbf{x} \in \mathbb{R}^3$ represents a point in 3D space. In signal analysis, average frequency or bandwidth is typically weighted by the power spectral density:

$$\hat{D}(\omega) = \sum_{i=1}^{n} o_i \hat{G}_i(\omega), \quad |\hat{D}(\omega)|^2 = \left|\sum_{i=1}^{n} o_i \hat{G}_i(\omega)\right|^2, \quad \bar{\omega} = \frac{\int_{-\infty}^{\infty} \omega\, |\hat{D}(\omega)|^2\, d\omega}{\int_{-\infty}^{\infty} |\hat{D}(\omega)|^2\, d\omega}, \tag{3}$$

where $\hat{D}(\omega)$ and $\hat{G}_i(\omega)$ denote the frequency-domain counterparts of $D(t)$ and $G_i(t)$. Prior to compute $\bar{w}$, we first compute the Fourier transform $F(\omega)$ of the Gaussian function $e^{-at^2}$ and it's half-power (3dB) bandwidth $\omega_{3dB}$ as: $F(\omega) = \sqrt{\frac{\pi}{a}} \exp\left(-\frac{\omega^2}{4a}\right)$ and $\omega_{3dB} = \sqrt{2a \ln 2}$. Detailed derivation is given in the Appendix. $\omega_{3dB}$ is commonly used to characterize the effective frequency range that contains the majority of the signal's energy, we use it to simplify the computation of the power spectrum $|\hat{D}(w)|^2$, since performing continuous integration over tens of millions of Gaussian primitives is computationally challenging.

Given that the frequency of each Gaussian primitive $\hat{G}_i(\omega) \propto F(\omega)$ follows a bell-shaped distribution, with its energy predominantly concentrated within $w_{3dB}$, we approximate the mean frequency of each primitive using $w_{3dB}$ and assume negligible intensity at other frequencies, the continuous-to-discrete approximation is shown in Fig. 3(b). Consequently, the $|\hat{D}(w)|^2$ can be approximated as:

$$|\hat{D}(\omega)|^2 = \sum_{i=1}^{n} |o_i \hat{G}_i(\omega)|^2 + \sum_{i \neq j} \text{Re}\left(o_i \hat{G}_i(\omega) \left(o_j \hat{G}_j(\omega)\right)^*\right) \approx \sum_{i=1}^{n} (2\pi)^3 \det(\boldsymbol{\Sigma_i}) o_i^2 \delta(\omega - \omega_{3dB_i}),$$

(4)

where $\text{Re}(\cdot)$ denotes the real part extraction, $\delta(\cdot)$ is the unit impulse function, and $\det(\cdot)$ denotes the determinant. As a result, we can estimate the average frequency of the entire scene as:

$$\bar{\omega} = \frac{\int_{-\infty}^{\infty} \omega \, |\hat{D}(\omega)|^2 \, d\omega}{\int_{-\infty}^{\infty} |\hat{D}(\omega)|^2 \, d\omega} = \frac{\sum_i^n o_i^2 \det(\boldsymbol{\Sigma_i}) \omega_{3dB_i}}{\sum_i^n o_i^2 \det(\boldsymbol{\Sigma_i})}.$$

(5)

Since for a 1D Gaussian function, $\omega_{3dB} = \sqrt{2a \ln 2} \propto \frac{1}{\sigma}$, we refer to the scale of a primitive as $\texttt{scale} = [\sigma_1, \sigma_2, \sigma_3]$ and adopt the average over the three axes as $\omega_{3dB_i}$: $\omega_{3dB_i} \propto \sum_{k=1}^{3} \frac{1}{3\sigma_k}$.

**Effectiveness Validation.** To identify the effectiveness of our definition of the average sample frequency and scene signal bandwidth, we conduct the following experiments using vanilla 3DGS (Kerbl et al., 2023):

(1) To validate the relation between sampling frequency (*i.e.*, image resolution in our perspective) and signal bandwidth, we reconstruct scenes at different image resolutions. For the relationship between $\omega_{3dB_i}$ and $\texttt{scale}$, we adopt three metrics: $\max \frac{1}{\sigma_k}$, $\min \frac{1}{\sigma_k}$ and $\sum_{k=1}^{3} \frac{1}{3\sigma_k}$. In Fig. 3(a), our definition of recovered scene bandwidth, with $\omega_{3dBi} \propto \sum_{k=1}^{3} \frac{1}{3\sigma_k}$, shows a clear positive correlation with the sampling frequency, whereas the remaining two lack such behavior. This arises as our focus is on the scene's average bandwidth, rather than isolated high or low frequencies. This strict positive correlation confirms our proposed definition's validity.

(2) We monitor the average scene bandwidth during training to validate the effectiveness of its definition. As shown in Fig. 3(a), under our definition, scene bandwidth increases steadily with training iterations, indicating a progressive recovery of higher-frequency signals.

### 3.3 COARSE-TO-FINE SIGNAL STRUCTURE RECOVERY

Based on the defined average sampling frequency and the effective scene bandwidth of the 3D Gaussian representation, we introduce a novel scheduler (SIG) that **S**ynchronizes **I**mage supervision with

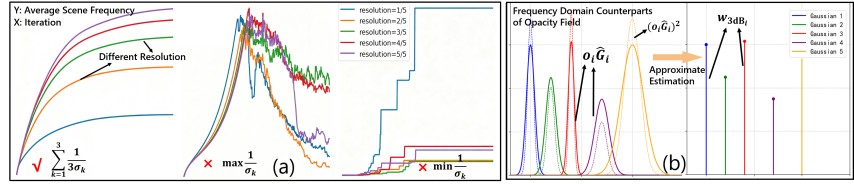

Figure 3: **Effectiveness Validation:** (a) Average Scene Bandwidth during Training under Different Image Resolutions and $w_{3dB}$ Estimation Methods. (b) Our approximation of the power spectrum.

Gaussian frequencies to adaptively switch resolutions and regulate densification accordingly. SIG consists of two key components: frequency-aligned resolution scheduler and densification scheduler.

**Frequency-Aligned Resolution Scheduler.** As shown in Eq. (1), computing the average sampling frequency requires integrating over differential elements of each scene patch. While the Nyquist sampling theorem is valid locally for each infinitesimal element, the global *average* value does not strictly satisfy it. Hence, we assess convergence at a given resolution by examining whether the average scene bandwidth stabilizes. Once convergence is achieved, we increase the resolution of training images for the next stage. Specifically, we compute the scene frequency $\omega_{iter}$ at each iteration using Eq. (5), and obtain its gradient as $\frac{d\omega}{d\text{iter}}$. The resolution is updated once the condition as Eq. (6) is satisfied, where $k$ is universal across scenes, $NN(\cdot)$ denotes nearest-neighbor search, which we use to normalize the scale, since the input point cloud may not correspond to real-world units.

$$\frac{d f}{d\,\text{iter}} = \omega_i - \omega_{i-1}, \quad d = NN(\text{point cloud}_{init}), \quad \frac{d f}{d\,\text{iter}} < k \cdot \text{mean}(\tfrac{1}{d}). \tag{6}$$

This strategy enables frequency-consistent signal recovery. As shown in Fig. 3(a), the resolution is increased as the scene approaches the maximum recoverable frequency.

**Densification Scheduler.** As noted earlier, increasing the sampling frequency requires more Gaussians to fit higher-frequency signals. Therefore, after each resolution update, we perform $m$ rounds of densification. This strategy avoids unconstrained over-densification in the early stage and prevents inefficient training caused by an excessive number of Gaussians.

## 3.4 STRUCTURE-AWARE OPTIMIZATION

**Sphere-Constrained Gaussians.** Due to sparse initialization, the Gaussian primitives exhibit an excessively large optimization space. Spatial-agnostic optimization of Gaussian positions often results in floaters. Scaffold-GS (Lu et al., 2024) employ neural anchors to decode Gaussians, which helps mitigate this issue. However, it prevents block-wise independent training due to the shared Gaussian decoder. In contrast, we constrain our explict Gaussians within a similar explicit structural framework. Specifically, during optimization, we assign each Gaussian ellipsoid two attributes: anchor and maximum offset. These attributes determine whether the current Gaussian deviates from the original geometric structure. Initially, all anchors are set to the initial points obtained from COLMAP (Schonberger & Frahm, 2016), and the offset is computed as $\mathbf{x}_{\text{current}} - \mathbf{x}_{\text{anchor}}$. The `max_offset` is determined based on the initialization by searching for the $K$ nearest neighbors of each point and calculating their average distance, which serves as the maximum offset. In practice, considering potential inaccuracies in COLMAP results but overall structural reliability, we relax the pruning criteria by introducing a scale factor $l > 1$ and setting $K = 15$. When the offset exceeds $l \times$ `max_offset`, the Gaussian point exceeding this spherical constraint is pruned.

**Anchor-based adaptive control.** Due to the introduction of anchors, newly generated Gaussians during densification are also assigned both `anchor` and `max_offset` attributes. The assignment rules as follows:

(1) *Replication.* Replication signals under-reconstruction—*i.e.*, the region exhibits low-frequency textures and is initially under-populated with Gaussians. Therefore, to allow large-scale spatial displacement, the newly created Gaussian does not inherit the original anchor. Its current position is set as the new anchor, while the `max_offset` attribute is inherited from the original Gaussian.
(2) *Splitting.* Splitting targets higher-frequency details. In this case, the anchor is inherited from the original Gaussian to preserve structural alignment. The `max_offset` is scaled down to $0.7\times$ of the original, encouraging more conservative updates and ensuring stability in high-frequency regions.
(3) *Densification Regularization.* Constraining Gaussians densification solely via 2D image supervision (*i.e.*, reconstruction loss) is inherently limited. We introduce regularization mechanisms specifically tailored for densification, based on reprojection-based photometric loss, which is commonly used in sparse-view reconstruction. Specifically, we extract dense point clouds from rendered depth maps $z$ and project them onto adjacent frame images to compute photometric errors. It is effective for large textureless regions, enforcing geometric-color consistency, and is calculated as Eq. (7):

$$\mathcal{L}_{\text{cons}} = \sum_{(i,j)} \|C_i\langle p\rangle) - C_j\langle \mathcal{F}_j(T_j T_i^{-1} \mathcal{F}^{-1}(z, p))\rangle\|_2^2, \tag{7}$$

where $T$ represents the camera pose. The $C\langle\cdot\rangle$ denotes the color obtained by sampling from the image. $\mathcal{F}(x)$ denotes the projection function that maps a 3D point $x$ onto a 2D image. $\mathcal{F}^{-1}(z, p)$

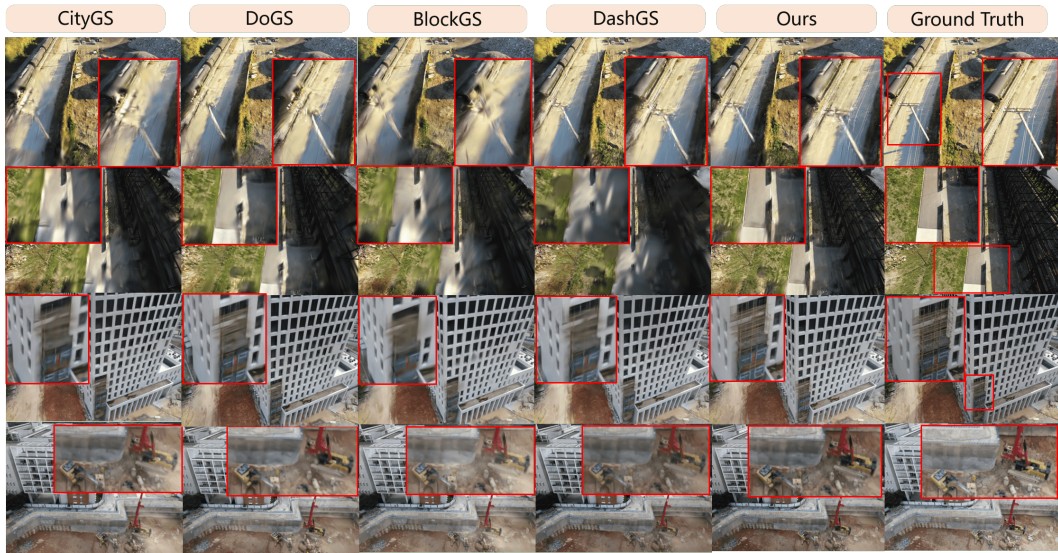

| CityGS | DoGS | BlockGS | DashGS | Ours | Ground Truth |

Figure 4: **Qualitative Results.** The top two rows illustrate sparse observations, whereas the bottom two represent regular regions. For regions corresponding to sparse viewpoints, our approach effectively minimizes the generation of redundant and erroneous Gaussian components. For general regions, our method recovers high-frequency details while maintaining intact geometric structures.

denotes the back-projection, mapping a pixel $p$ to a 3D point using depth $z$. We found that using it throughout the entire optimization does not improve accuracy. This indicates that GS-based depth estimation is not perfectly precise, which makes the projections inexact. However, it can still provide useful constraints during densification, serving as a densification regularization.

## 4 EXPERIMENTS

### 4.1 EXPERIMENTS SET UP

**Datasets and Metrics.** Following large-scale reconstruction methods (Liu et al., 2024; Wu et al., 2025), we evaluate our approach on three datasets: real-world Mill19 (Turki et al., 2022), UrbanScene3D (Lin et al., 2022), and synthetic MatrixCity (Li et al., 2023). We report results using metrics: SSIM, PSNR, and LPIPS (Zhang et al., 2018). A color correction is applied for metric computation, following (Lin et al., 2024).

**Implementation Details.** All images are downsampled by a factor of $n$ (from 5 to 1) to enable dynamic resolution. All methods are trained and evaluated on NVIDIA RTX 4090 GPUs. The resolution scheduler threshold for $\frac{df}{d\text{iter}}$ is set to $k \cdot \text{mean}(\frac{1}{d})$ with $k = 5 \times 10^{-5}$, and $\frac{df}{d\text{iter}}$ is evaluated every 100 iterations. For Sphere-Constrained Gaussians, we adopt a scale factor of $l = 15$. We use 30,000 training iterations for both the coarse and fine stages. More details can be found in Appendix.

### 4.2 COMPARISON WITH OTHER METHODS.

**Baselines.** Our method is benchmarked against both NeRF-based and GS-based approaches. The NeRF-based baselines include Mega-NeRF (Turki et al., 2022) and SwitchNeRF (Zhenxing & Xu, 2022), whereas the 3DGS-based methods comprise VastGS (Lin et al., 2024), CityGS (Liu et al., 2024), DOGS (Chen & Lee, 2025), and BlockGS (Wu et al., 2025). Additionally, we compare our method with DashGS (Chen et al., 2025)—a hard-coded scheduling strategy. More details can be found in Appendix.

**Reconstruction Quality.** Our method implements structure-aware optimization, enabling the fitting of high-frequency details with more Gaussian primitives while avoiding floaters. Consequently, we can control the number of Gaussians by adjusting the gradient threshold during densification, lead-

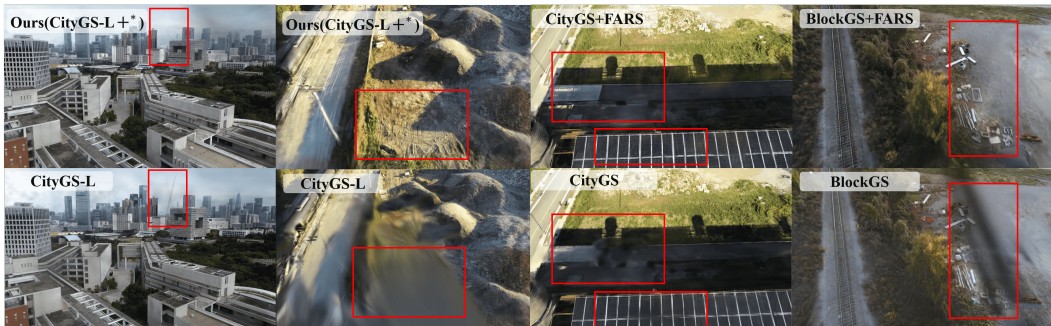

Figure 5: **Ablation.** CityGS-L: lower densification threshold of CityGS for higher-frequency fitting.

Table 1: **Quantitative comparison of NVS results.** The best, the second best, and the third best results are highlighted in red , orange and yellow .

| Scenes | building | | | rubble | | | residence | | | sci-art | | | MatrixCity-Aerial | | |
|---|---|---|---|---|---|---|---|---|---|---|---|---|---|---|---|
| | PSNR | SSIM | LPIPS | PSNR | SSIM | LPIPS | PSNR | SSIM | LPIPS | PSNR | SSIM | LPIPS | PSNR | SSIM | LPIPS |
| Mega-NeRF | 20.92 | 0.547 | 0.454 | 24.06 | 0.553 | 0.508 | 22.08 | 0.628 | 0.401 | 25.60 | 0.770 | 0.312 | - | - | - |
| Switch-NeRF | 21.54 | 0.579 | 0.397 | 24.31 | 0.562 | 0.478 | 22.57 | 0.654 | 0.352 | 26.51 | 0.795 | 0.271 | - | - | - |
| DashGS | 22.65 | 0.770 | 0.251 | 26.37 | 0.802 | 0.237 | 23.40 | 0.825 | 0.210 | 24.10 | 0.833 | 0.239 | 28.66 | 0.869 | 0.203 |
| VastGS | 21.80 | 0.728 | 0.225 | 25.20 | 0.742 | 0.264 | 21.01 | 0.699 | 0.261 | 22.64 | 0.761 | 0.261 | 28.33 | 0.835 | 0.220 |
| CityGS | 22.70 | 0.774 | 0.246 | 26.45 | 0.809 | 0.232 | 23.35 | 0.822 | 0.211 | 24.49 | 0.843 | 0.232 | 28.61 | 0.868 | 0.205 |
| DOGS | 22.73 | 0.759 | 0.204 | 25.78 | 0.765 | 0.257 | 21.94 | 0.74 | 0.244 | 24.42 | 0.804 | 0.219 | 28.58 | 0.847 | 0.219 |
| BlockGS | 21.11 | 0.750 | 0.234 | 25.52 | 0.801 | 0.233 | 22.15 | 0.810 | 0.211 | 24.18 | 0.831 | 0.219 | 28.17 | 0.863 | 0.199 |
| **Ours-L** | 22.86 | 0.778 | 0.225 | 27.35 | 0.843 | 0.189 | 23.46 | 0.840 | 0.194 | 25.94 | 0.890 | 0.170 | 29.04 | 0.888 | 0.184 |
| **Ours-S** | 22.98 | 0.780 | 0.225 | 27.21 | 0.830 | 0.208 | 23.50 | 0.839 | 0.196 | 25.12 | 0.863 | 0.200 | 29.01 | 0.879 | 0.181 |

ing to two model variants with different Gaussian counts—where the model with more Gaussians (Ours-L) demonstrates the higher performance ceiling of our algorithm. Besides, for BlockGS, We follow its official implementation and use a batch size of 4 for better reconstruction quality (processing 4 images per iteration to enhance results, albeit with proportional time overhead). Please refer to Appendix for details of BlockGS. As shown in Table 1 and Table 3, our method achieves comprehensive improvements over baseline approaches, such as a +0.9 dB gain in PSNR for the scene 'rubble'. Fig. 4 illustrates that compared to baselines, our method captures finer high-frequency details without producing floaters (NeRF-based results can be found in Appendix. Higher SSIM and LPIPS scores further validate its superior visual quality. Furthermore, while DashGS yields modest gains over baselines via its specialized training strategy, its ability to represent high-frequency information in large-scale scenes remains limited, as its design prioritizes training efficiency.

**Efficiency Analysis.** Considering that different block partitioning strategies, as well as the size and number of blocks, affect computational time, we select two baselines corresponding to two partitioning strategies and report the average optimization time per block on scene 'rubble' in Table 3. Our method substantially improves training efficiency, achieving $1.5\times$ and $1.4\times$ speedups on BlockGS and CityGS, respectively.

## 4.3 ABLATION STUDY

As shown in Fig. 3, we have validated the effectiveness of our definitions for the average sampling frequency and Gaussian representation frequency. In this section, we conduct ablation studies on the 'rubble' to verify the efficacy of each individual module, with results presented in Table 2.

**Frequency-Aligned Resolution Scheduler (w/o FARS).** We directly train on the highest-resolution images and, following CityGS, perform densification every 200 iterations between 500 and 15,000 iterations. Results show that omitting FARS leads to a significant drop in reconstruction quality and increased training time. Due to the lower splitting threshold, over-densification even yields results inferior to CityGS. Furthermore, as illustrated in Fig. 5 and Table 2 (1.5M points vs. 2.2M points), without dynamic resolution scheduling, redundant Gaussians and floaters tend to emerge in the early stages of optimization, particularly when the gradient threshold for densification is set low.

Table 2: **Ablation.** Frequency-consistent training yields substantial performance gains.

| Method | PNSR | SSIM | LPIPS | Points |
|---|---|---|---|---|
| w/o-FARS | **26.18** | **0.807** | **0.231** | **2.2M** |
| w/o-DS | 26.89 | 0.827 | 0.212 | 1.4M |
| w/o-SCG | 27.05 | 0.818 | 0.210 | 1.8M |
| w/o-DR | 27.01 | 0.820 | 0.209 | 1.9M |
| w/ all(*) | **27.35** | **0.843** | **0.189** | **1.5M** |

Table 3: **Integration with other baselines.** We improve quality and efficiency by augmenting baselines with our method (*). Opt.time: min/block.

| Method | PNSR | SSIM | LPIPS | Opt |
|---|---|---|---|---|
| BlockGS | 25.52 | 0.801 | 0.233 | 201 |
| BlockGS+* | 25.89 | 0.820 | 0.228 | 130 |
| CityGS | 26.45 | 0.809 | 0.232 | 98 |
| CityGS+* | 27.35 | 0.843 | 0.189 | 71 |

**Densification Scheduler (w/o DS).** Building on the "w/o FARS" setup, we adopt dynamic resolution scheduling while retaining the original splitting strategy. This leads to a potential issue: if the moment of reaching the highest resolution lags behind the termination of densification, no further Gaussian densification occurs afterward. Such a mismatch impairs the model's capacity to represent high-frequency information.

**Sphere-Constrained Gaussians (w/o SCG) and Densification Regularization (w/o DR).** Sphere-Constrained Gaussians (SCG) explicitly leverages the geometric structure of initialized sparse point clouds, thereby reducing the probability of Gaussians being optimized into erroneous regions. As for the Densification Regularization, in the absence of ground-truth depth, this method establishes geometric consistency constraints across multiple frames. Once floaters emerge, they disrupt depth prediction and thereby induce errors in adjacent views. As shown in Table 2, both methods effectively reduce the number of Gaussians that suffer from redundancy or erroneous optimization (1.9M points vs. 1.5M points).

**Integration with Other Baselines.** Notably, our strategy is compatible with most baseline methods and can be used as a plug-and-play component. As shown in Table 3, our approach significantly improves both rendering quality and efficiency when applied to BlockGS and CityGS.

## 5 LIMITATION

In large-scale scenes, Level-of-Detail (LoD) rendering strategy is usually required. While our approach can leverage existing LoD strategies such as those proposed in CityGS, our coarse-to-fine training naturally produces both coarse and fine Gaussians, providing a more intrinsic way of constructing hierarchical structures. This also points to a promising direction for future optimization.

## 6 CONCLUSION

We attribute low-quality reconstruction under sparse point cloud initialization, floaters, and redundant Gaussian primitives to frequency misalignment between supervision and target signal. Based on this, we derive the average scene and supervision frequencies and present a novel training framework. We introduce a unified resolution–densification scheduling strategy driven by scene frequency convergence, and propose Sphere-Constrained Gaussians to leverage initial geometry while regularizing densification. This enables frequency-consistent and geometry-aware optimization.

## 7 ACKNOWLEDGEMENT

This work was supported by the National Key Research and Development Program of China (No. 2024YFE0211000), in part by the National Natural Science Foundation of China (No. 62372329, 62506263, 62506264), in part by the Shanghai Scientific Innovation Foundation (No. 23DZ1203400), in part by the Fundamental Research Funds for the Central Universities, in part by the Key Technology Development and Integrated Application of Guided Autonomous Vehicles Project, in part by the China Postdoctoral Science Foundation (No. BX20250383, GZB20250385, 2025M771530, 2025M771539), in part by Tongji-Qomolo Autonomous Driving Commercial Vehicle Joint Lab Project, and in part by Xiaomi Young Talents Program.

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
