# OpenReview forum: "Signal Structure-Aware Gaussian Splatting for Large-Scale Scene Reconstruction"
_ICLR.cc/2026/Conference — ICLR 2026 Poster_

### Official Review · Reviewer_aMuM · 2025-10-25

**Soundness:** 4
**Presentation:** 2
**Contribution:** 3
**Rating:** 8
**Confidence:** 4

**Summary:**

This paper identifies a key problem in scaling 3D Gaussian Splatting to large-scale scenes: the mismatch between sparse initial 3D points and high-frequency image supervision. This mismatch causes uncontrolled growth of redundant Gaussians, hurting both efficiency and quality. The authors reframe the problem as one of signal frequency synchronization and propose a scheduler to align the 2D image supervision with the 3D Gaussian representation's evolving complexity.

**Strengths:**

1. The sufficient experiments adequately illustrate the superiority of the method and the effectiveness of each model design. The settings of experiments and baseline implementations are also well explained, making it easier to follow the paper and reproduce results.
2. The efficiency and quality of large-scale scene reconstruction are of significance. This paper introduces a sound and innovative method to push the performance bound over existing methods.

**Weaknesses:**

1. The order of LPIPS and SSIM in Tab.2 and Tab.3 seems to be wrong.
2. Fig.2 itself is not well self-contained. The formulas and pipelines are, to some extent, confusing, such as the meaning of the blue dashed line. And it should provide a reference to the related sections.

**Questions:**

See weakness.

---

> ### Author Response · Authors · 2025-11-21
> **Response to Reviewer aMuM - Stage 1**
>
> Thank you sincerely for your careful and detailed review. We have thoroughly addressed the concerns you raised, and your suggestions have improved the clarity and overall presentation of our paper. A detailed response to each concern is provided below.
>
> **Clarify Figure 2 and Incorrect Metric Order.**
>
> Regarding the **Incorrect Metric Order in Tables**, we noticed the order of SSIM and LPIPS was incorrect in Table 2,3 and sincerely appreciate your careful review. These issues have been corrected in the revised version.
>
> Moreover, as **Figure 2** presents the overall pipeline, ensuring its completeness and clarity is crucial. In the original version, our intention was that the upper part illustrates the resolution scheduler, while the lower part shows the densification scheduler, with the middle portion highlighting how the two schedulers are frequency-aligned. However, we noticed that the blue dashed line conveyed mixed meanings: on the one hand, it attempted to explain the Frequency Matching Module; on the other hand, it pointed to the supervision and recovery frequency schedules. This caused ambiguity.
>
> Following your suggestion, we have removed the unclear connecting lines and added explicit annotations to improve interpretability. Since detailed formulas are unnecessary in this figure, we now use the simplified labels “Sampling frequency” and “Recovery signal frequency” for clarity. In addition, in the latest version, each module in the figure will be clearly linked to its corresponding section in the paper.
>
> We sincerely thank you again for your recognition, careful review, and valuable suggestions.

---

### Official Review · Reviewer_P2B2 · 2025-10-26

**Soundness:** 3
**Presentation:** 3
**Contribution:** 3
**Rating:** 6
**Confidence:** 2

**Summary:**

This paper proposes a frequency-consistent framework for large-scale 3D scene reconstruction. It mathematically defines the average sampling frequency and signal bandwidth of 3D Gaussian representations, and introduces a scheduler that adaptively adjusts image resolution and Gaussian densification based on scene frequency convergence. Additionally, it proposes Sphere-Constrained Gaussians to enforce geometry-aware optimization and reduce redundancy. The paper achieves state-of-the-art rendering quality and faster training compared to the baselines.

**Strengths:**

- This paper formalizes when to raise resolution and when to densify using a bandwidth convergence signal, which improves the performances of baselines that rely on predefined schedules
- The proposed method is a plug-and-play component that could be applied to multiple baselines (e.g., 3DGS, CityGS, BlockGS) and leads to consistent performance gains and better efficiency
- The ablations studies clearly show the benefits brought by each of the proposed components

**Weaknesses:**

- The convergence threshold k, neighbor count K, and scaling factor l (e.g., K=15, l=15, and max_offset is scaled down to 0.7×) are stated but not sufficiently justified. Are these values chosen empirically, or derived from any principled analysis? Moreover, do they require dataset-specific tuning, especially under varying levels of SfM sparsity or reconstruction noise?
- Missing baselines: some relavant works in this domain are not compared to, especially: CityGaussianV2 (ICLR 2025), FlashGS (CVPR 2025)

**Questions:**

As noted in the weaknesses, I recommend that the authors include ablations on key hyperparameters, or at least provide a discussion on how these values are selected and whether they are kept consistent across all experiments. I also suggest to provide additional qualitative/quantitative comparisons with CityGaussianV2 and FlashGS.

Apart from above, I have an additional question:
- Does the proposed method also improve geometry? Although it's hard to directly evaluate geometries in such large scenes, I would be interested to see a similar geometry evaluation as in CityGaussianV2.

---

> ### Author Response · Authors · 2025-11-21
> **Response to Reviewer P2B2 - Stage 1**
>
> We sincerely appreciate your detailed review and recognition. Your comments are highly valuable and help to strengthen the clarity and completeness of our method. Below, we provide our responses to your concerns.
>
> **W1: Hyperparameter Sensitivity and Selection.**
>
> We agree that hyperparameter selection is crucial for the practical applicability of the method. Below, we provide a detailed explanation and analysis of the rationale behind these choices and their sensitivity.
>
> (1)**Convergence threshold k**: We increase the supervision resolution when
> $ \frac{df}{d\text{iter}} < k $.
> Since different scenes may have inconsistent scales (because COLMAP initialization is not necessarily aligned with real-world scale), we normalize the threshold using
> $ \mathrm{mean}(1/d) $.
> This normalization makes the scales across scenes consistent, ensuring that the threshold $ k $ is independent of the scene scale. We use the same convergence parameter across all large-scale datasets, as
> $ k = 5 \times 10^{-5} $
> works consistently for all five large-scale scenes in the three datasets.
>
> As for **Sensitivity and Selection Principles**, our choice of $k$ follows the principle that it should ensure convergence to the highest frequency before training ends. We observe that the iteration at which the frequency converges varies significantly across datasets. For example, when training at 1/5 resolution on "Rubble", the first Frequency-Aligned state is reached after around 4200 iterations, while in Sci-Art this occurs at about 7800 iterations. This variation arises from the uncertain relationship between the initialized point cloud and the supervisory sampling frequencies. However, as training progresses, the discrepancy between the supervision signal and the reconstructed signal gradually diminishes. We find that all scenes eventually converge to the highest resolution at around 20k iterations.
>
> We also experimented on Rubble with $k = 7 \times 10^{-5}$ and $k = 3 \times 10^{-5}$, and illustrate how different thresholds influence the resolution adjustment. Overall, within a certain range, these variations have little effect on the final evaluation results. (We limit the computation of the frequency to once every 100 iterations, and it is updated only after the frequency change rate stabilizes below $100 \times k$). The table below shows the iteration numbers for different resolution transitions and the final evaluation metrics:
>
> |($\times 10^{-5}$)|1/5 | 2/5  | 3/5  | 4/5   | 5/5   | PSNR  | SSIM  | LPIPS |
> |-----|------|------|-------|-------|-------|-------|-------|-------|
> | 3   | 0    | 2700 | 8100  | 16200 | 18500 | 27.22 | 0.844 | 0.188 |
> | 7   | 0    | 7400 | 9000  | 17500 | 23000 | 27.31 | 0.839 | 0.192 |
> | 5   | 0    | 4200 | 8900  | 17000 | 21000 | 27.35 | 0.843 | 0.189 |
>
>
> *(1/5–5/5 denotes resolutions relative to the final output size. The columns labeled 1/5–5/5 indicate the iteration numbers at which the training transitions to the corresponding resolution.)*
>
> (2) The second key set of hyperparameters is in the **Structure-Aware Optimization**. We explicitly define the optimization bounds for each Gaussian to reduce the occurrence of floaters: $l$ times the average distance of the $K$ nearest neighbors for each Gaussian, i.e., $l \cdot \text{NN}(K)$, where NN(K) denotes the K-nearest neighbor search. Here, NN(K) reflects the density of Gaussians within a local patch around each point. Since $K$ and $l$ have similar effects, $K$ only needs to be consistent across different regions and datasets, whereas $l$ serves as the tunable parameter. Our choice of $l = 15$ is based on the principle that the number of Gaussians pruned in the early stage should generally be less than the number of Gaussians added through densification, ensuring that the scene frequency can increase steadily. We set $l$ to a relatively large value to filter out significantly deviating Gaussians. The results show that $l = 15$ works well across five scenes in three datasets. During the early stages of optimization, when the learning rate is high, these geometric constraints play a crucial role. For example, in Rubble, with an initial count of 1,694,315 Gaussians, the pruning step in the splitting stage removes several hundred to several thousand Gaussians in the early phase. Later in the optimization, this number drops to the order of tens, and regardless of the value of $l$, the final number of pruned Gaussians stabilizes to the order of tens as optimization progresses. We tried different combinations on "Rubble", and the results are shown below.
>
> | l  | K  | final points | PSNR  | SSIM  | LPIPS |
> |----|----|--------------|-------|-------|-------|
> | 5  | 15 | 1.4M         | 27.30 | 0.841 | 0.190 |
> | 15 | 15 | 1.5M         | 27.35 | 0.843 | 0.189 |
> | 15 | 5  | 1.4M         | 27.29 | 0.843 | 0.192 |

---

> ### Author Response · Authors · 2025-11-21
> **Response to Reviewer P2B2 - Stage 2**
>
> (Continuing from W1)
>
> Additionally, during Gaussian densification, the maximum offset is scaled down to $0.7\times$ (less than 1). The rationale is that densification aims to reconstruct higher-frequency signals, so the optimization space should be further constrained to avoid generating floaters. In our experiments, we found that this parameter has little effect within a certain range, and thus we fix it at $0.7\times$ across all datasets and scenes.
>
> In summary, our method does not require dataset-specific tuning, and all hyperparameters can be directly applied across different datasets. It is robust to varying levels of SfM sparsity. This is partly due to normalization using $\mathrm{mean}(1/d)$ and $\text{NN}(K)$, and partly because varying SfM sparsity mainly affects pruning and scheduling during the early stages of optimization, but these effects stabilize over time and, within a reasonable range, do not significantly affect the final results.
>
> **W2: Expanded Baseline Comparison.**
>
> We sincerely appreciate your insightful suggestions, which substantially contribute to improving the quality of our work. In the initial version of the manuscript, our methodology mainly followed CityGSv1, BlockGS, and other 3DGS-based approaches, which primarily focus on image-level reconstruction quality. Following your suggestion, we conducted comparative experiments with CityGSv2[1], which serves as a representative baseline for large-scale scene geometric reconstruction.
>
> We selected the "Rubble"(Rb) from Mill19 and the MatrixCity-Aerial (Mc) for evaluation. Since CityGSv2 mainly focuses on surface reconstruction, its objectives are not fully aligned with those of 3DGS-based methods, which emphasize image-level rendering quality. Consequently, CityGSv2 performs less favorably on standard image reconstruction metrics such as PSNR, SSIM, and LPIPS  (*Qualitative results are provided in Appendix.7 in the updated version*). Regarding geometry-oriented metrics, we provide a detailed discussion in the next question.
>
>
> || **Mc**  | |  |**Rb**|  |  |
> |-----------|-------|-------|-------|-----------|-------|-------|
> | Method    | PSNR  | SSIM  | LPIPS |   PSNR  | SSIM  | LPIPS |
> | CityGSv1  | 28.61 | 0.868 | 0.205 |  26.45 | 0.809 | 0.232 |
> | CityGSv2  | 28.32 | 0.858 | 0.169 |  26.03 | 0.800 | 0.229 |
> | Ours      | 29.01 | 0.879 | 0.181 |  27.35 | 0.843 | 0.189 |
>
> *(Note: The PSNR of CityGSv2 in our experiments is slightly better than those reported in the original paper, because all methods in our evaluation pipeline undergo consistent color correction, following the procedures described in DOGS and BlockGS.)*
>
> Regarding FlashGS[2], we observe that it is a highly effective tool for accelerating 3DGS inference, particularly for high-resolution images, while maintaining rendering quality. Following the official implementation, FlashGS is specifically designed for inference acceleration rather than model training (it currently supports only forward computation and does not support gradient backpropagation).
>
> Taking your suggestion into account, we applied FlashGS to render the results of our trained models. Using the official setup, we evaluated our models on MatrixCity and "Rubble". The results demonstrate that FlashGS achieves the same rendering quality while improving rendering efficiency. We also recognize that acceleration is a crucial direction for 3DGS. In the updated version of our paper, we discussed various acceleration strategies and cited influential works such as FlashGS. Integrating these acceleration techniques with our model has the potential to further enhance its efficiency.

---

> ### Author Response · Authors · 2025-11-21
> **Response to Reviewer P2B2 - Stage 3**
>
> **Q1: Geometry Evaluation**
>
> Compared with CityGSv1, our method is able to improve geometry, as we can generate more Gaussians constrained by geometric priors and effectively reject floaters. As 3DGS-based methods primarily focus on image reconstruction quality, the resulting geometric accuracy is still not as strong as that achieved by 2DGS-based methods such as CityGSv2. To assess geometric performance, we conduct the same geometry evaluation on the MatrixCity dataset following CityGSv2, which provides ground-truth point clouds. The results are shown below.
>
> **MatrixCity-Aerial: Geometric Metrics**
>
> | Method    | Precision | Recall | F1    |
> |-----------|-------|-------|-------|
> | CityGSv1  | 0.362 | 0.637 | 0.462 |
> | CityGSv2  | 0.441 | 0.752 | 0.556 |
> | Ours      | 0.401 | 0.644 | 0.494 |
>
> As can be seen, compared with the 3DGS-based method CityGSv1, our approach achieves improved geometric metrics on the MatrixCity-Aerial scene. In addition, for the Rubble scene, we conducted an additional evaluation. We used the sparse point cloud generated by the original COLMAP reconstruction as the ground-truth point cloud, and extracted the positions of the final Gaussians as the predicted point cloud. Based on these two point sets, we computed the Chamfer Distance (CD), which provides a measure of how much the model deviates from the original underlying geometry.
>
> **Rubble: Chamfer Distance**
>
> | Method    | CD    |
> |-----------|-------|
> | CityGSv1  | 0.177 |
> | CityGSv2  | 0.142 |
> | Ours      | 0.137 |
>
> When computing the Chamfer Distance (CD), it is important to account for the inevitable noise present in both the SFM point cloud and the point cloud generated from Gaussians. Since the SFM point cloud of the Rubble scene is represented at real-world scale, certain noisy correspondences may exhibit distances on the order of hundreds of meters. To avoid such outliers from dominating the metric, we discard any point pairs whose distance exceeds 5 meters.
>
> Moreover, the derivations from Eq. 2 to Eq. 5 are equally applicable to 2DGS, and our method is plug-and-play. In principle, it can be extended to both 2DGS and CityGSv2, and in future work, we plan to further investigate this extension.
>
> We would like to express our sincere gratitude once again for your insightful comments and valuable questions.
> Besides, we will incorporate the above supplementary content into the main text and appendix, with additional qualitative results also provided in the appendix.
>
> [1] Liu, Yang, et al. "Citygaussianv2: Efficient and geometrically accurate reconstruction for large-scale scenes." arXiv preprint arXiv:2411.00771 (2024).
>
> [2] Feng, Guofeng, et al. "Flashgs: Efficient 3d gaussian splatting for large-scale and high-resolution rendering." Proceedings of the Computer Vision and Pattern Recognition Conference. 2025.

---

### Official Review · Reviewer_bS7H · 2025-10-29

**Soundness:** 2
**Presentation:** 3
**Contribution:** 3
**Rating:** 6
**Confidence:** 4

**Summary:**

This paper aims at solving the frequency inconsistent issue during 3DGS training. It leverages the frequency information to check whether it is necessary to increases the image resolution during training. Moreover, it introduces the Sphere-Constrained Gaussians to constrain its moving offset. Experiments under large-scale dataset validates its effectiveness since it beats SOTA large-scale 3DGS approaches.

**Strengths:**

(1) The idea is novel, which leverage the frequency information to schedule the resolution of training images.

(2) The writing is good and the paper is easy to understand.

(3) The proposed method is effective, and beats SOTA methods in large-scale datasets.

**Weaknesses:**

(1) The method is compared to many SOTA methods, but it lacks comparison with a naive baseline: scheduling the image resolutions during training at some hard-coded iterations (e.g. 10,000, 15,000, 20,000), I think some methods did this but it needs to be done in the experiment of this paper to provide a direct comparison.

(2) The experiments part lacks comparison on some standard benchmarks, e.g. the mipnerf360 dataset and the tanks-and-temples dataset.

**Questions:**

(1) Eq. (5) and Eq.(6) are computed at each training iteration. Will that be time consuming?

(2) At line327, the author mentioned "we use 30,000 training iterations for both the coarse and fine stages". How to define the coarse stage and fine stage? Since the method enlarges the resolution of training images in a coarse-to-fine manner, it is confused to me the 3DGS are trained with a coarse stage and a fine stage?

---

> ### Author Response · Authors · 2025-11-21
> **Response to Reviewer bS7H - Stage 1**
>
> We sincerely thank you for your careful reading and valuable suggestions, which help to improve and strengthen the quality of our work. In response to your concerns, the detailed explanations and additional information are provided below:
>
> **W1: Comparison with some hard-coded methods.**
>
> We sincerely appreciate your valuable suggestions, which substantially improved the completeness of our work. Based on your feedback, we constructed the following baseline. We use the same block partitioning and coarse initialization as our method. During block-wise training, the image resolution is divided into five levels (1/5, 2/5, 3/5, 4/5, and full), and these levels are linearly mapped to the training iterations (0–6000, 6001–12,000, etc.). For densification, we evaluate two strategies. The first follows CityGS[1], performing densification only in the 500-15,000 iterations. The second performs 3,000 densification iterations at each resolution level. On the scene "Rubble", the first strategy yields poor results because Gaussians split only at low resolutions and stop splitting once full resolution is reached, causing a mismatch between the densification and resolution schedulers.The second strategy still suffers from misalignment because it fails to account for the scene’s frequency characteristics, and its final performance is also inferior to the vanilla CityGS. In addition, we include DashGS[1] as a baseline in our main text (**the first row** of Table 1), which also employs a scheduler that linearly maps different image frequencies in the 2D color space to the training iterations. The results are as follows:
>
> | Method      | PSNR  | SSIM  | LPIPS | Points |
> |------------|-------|-------|-------|--------|
> | Strategy-1 | 26.21 | 0.804 | 0.240 | 1.4M   |
> | Strategy-2 | 26.39 | 0.811 | 0.240 | 1.6M   |
> | DashGS     | 26.37 | 0.802 | 0.237 | 1.3M   |
> | Ours       | 27.35 | 0.843 | 0.189 | 1.5M   |
>
> **W2: Evaluation on additional standard benchmarks.**
>
> We sincerely appreciate your insightful suggestion. A robust method should indeed generalize across different environment settings. Although our scheduler was originally designed for large-scale scene reconstruction, our approach remains effective on small-scale scenes as well. Following your recommendation, we conducted additional experiments on the Mip-NeRF360[2] and Tanks-and-Temples[3] datasets.
>
> For small-scale scenes, the divide-and-conquer strategy used in large-scale scenarios is unnecessary. Because our method is fully plug-and-play, we use the original vanilla 3DGS[4] as the baseline and report 3DGS + ours as our method, and all training configurations strictly follow the original 3DGS settings without any modification. In addition, our hyperparameters remain identical to those used for large-scale scenes.
>
> Surprisingly, although our method was initially designed for large-scale reconstruction, it still significantly improves both the performance and efficiency of vanilla 3DGS on small-scale datasets. Our approach achieves nearly 2× training speedup, and outperforms methods of DashGS and vanilla 3DGS. On the Tanks-and-Temples dataset, we evaluate on the Train and Truck scenes; on the Mip-NeRF360 dataset, we evaluate on the Flowers and Bicycle scenes. The results are as follows (**Qualitative results are provided in Appendix 6 in the updated version**):
>
>
>
> |**Tanks-and-Temples**         | Train |      |       | Truck |      |       |
> |---------|-------|------|-------|-------|------|-------|
> | Method  | PSNR  | SSIM | LPIPS | PSNR  | SSIM | LPIPS |
> | 3DGS    | 21.09 | 0.80 | 0.22  | 25.19 | 0.88 | 0.15  |
> | DashGS  | 22.01 | 0.80 | 0.22  | 25.66 | 0.88 | 0.16  |
> | Ours    | 22.33 | 0.82 | 0.20  | 25.90 | 0.88 | 0.15  |
>
> | **Mip-NeRF360**        | Flowers |      |       | Bicycle |      |       |
> |---------|-------|------|-------|-------|------|-------|
> | Method  | PSNR  | SSIM | LPIPS | PSNR  | SSIM | LPIPS |
> | 3DGS    | 21.52 | 0.61 | 0.34  | 25.25  | 0.77 | 0.21  |
> | DashGS  | 22.17 | 0.63 | 0.32  | 25.70  | 0.78 | 0.20  |
> | Ours    | 22.20 | 0.64 | 0.33  | 25.77  | 0.78 | 0.21  |
>
> Although our method was originally motivated by the significant mismatch between the frequencies of the sparsely initialized Gaussians and the sampling frequency in large-scale scenes, this phenomenon also persists in small-scale scenes. This occurs because, while the SFM-initialized point cloud in small-scale scenes indeed exhibits higher initial frequencies, the substantially higher sampling density leads to a correspondingly larger sampling frequency. For example, in the truck scene from Tanks and Temples, the average Gaussian frequency increases from 9.07 at initialization to 35.92 after convergence (compared with 3.23 to 14.35 in the large-scale scene rubble). Therefore, the frequency-aligned strategy remains effective even in small-scale scenarios.

---

> ### Author Response · Authors · 2025-11-21
> **Response to Reviewer bS7H - Stage 2**
>
> **Q1: Training-time efficiency concern.**
>
> In practice, this computation is extremely cheap and incurs negligible memory or runtime overhead. As shown in Eq.5, $w_{3\mathrm{dB}}$ and the opacity are obtained directly from the Gaussian parameters, but the weight also involves the determinant of the covariance, which may be your main concern. Although the determinant must be computed, it adds almost no computational burden. This is because 3DGS explicitly parameterizes the Gaussian ellipsoid by its three axis scales rather than by the full covariance matrix. Consequently, the determinant can be obtained directly from the product of the axis scales. A brief derivation is as follows.
>
> Let $\Sigma$ be the covariance matrix and let its eigen-decomposition be $\Sigma = Q \Lambda Q^\top$ with $\Lambda=\mathrm{diag}(\lambda_1,\lambda_2,\lambda_3)$. For a Mahalanobis ellipsoid defined by
> $(x-\mu)^\top \Sigma^{-1}(x-\mu) = c^2,$ the semi-axis lengths are $a_i = c\sqrt{\lambda_i}$. If we denote the ellipsoid semi-axes by $s_i$ (so $s_i = a_i$), then $\lambda_i = (s_i / c)^2$. Hence $ \det(\Sigma) = \prod_{i=1}^3 \lambda_i = \prod_{i=1}^3 \left(\frac{s_i}{c}\right)^2 = \frac{(s_1 s_2 s_3)^2}{c^{6}}. $
> Therefore, the determinant of the covariance is proportional to the square of the product of the three axis lengths, and we can directly compute the determinant using the Gaussian parameters: $\det(\Sigma) \propto (s_1 s_2 s_3)^2.$
>
> We can perform a simple estimation: assuming there are 2 million Gaussian points, the computation of Eq.5 takes approximately 0.12 ms per iteration (tested on NVIDIA GeForce RTX 3090). Since we compute this every 100 iterations, the total time overhead is  $0.12 \text{ms} \times \frac{30,000}{100} \approx 36\text{ms}.$
>
> For Eq.6, the `max offset` is computed only once at initialization. Subsequent computations merely involve calculating the difference to obtain the offset and comparing it with `max offset`. Therefore, the computational cost of this step is negligible. Finally, this demonstrates that the overhead introduced by these operations is minimal.
>
> **Q2: Question about the definition of coarse and fine stages.**
>
> In large-scale scene reconstruction, to reduce GPU memory usage and improve rendering efficiency, a divide-and-conquer strategy is typically employed. Specifically, Gaussian primitives are partitioned into blocks for parallel training, and corresponding supervision images are assigned to each block.
>
> To obtain a reliable partition of the primitives, most pipelines—such as CityGS[5]—require a pre-training (coarse training) stage to first optimize a scaffold. This scaffold is then used to partition the scene into blocks based on the initial rendering results (SSIM loss).
> Methods such as BlockGS[6] do not perform pre-training, instead, they partition blocks purely based on spatial relations and assign images based on visibility. We find that this direct assignment is less effective than coarse-training–based approaches. Consequently, we adopt the block partitioning strategy of CityGS, which includes a coarse training stage to initialize both the scaffold and the Gaussian primitives, and partitions blocks using SSIM loss in conjunction with block–image spatial relations. The optimization in our coarse-training stage follows Hierarchical 3DGS[7]: it makes only minimal adjustments to Gaussian positions, disables densification, and can be performed using low-resolution images to further accelerate the process.
> During fine training, each block is further optimized using our frequency-aligned schedule from low to high resolution. Since our method is orthogonal to block partitioning strategies, it can also be applied to approaches like BlockGS that do not require coarse training.
>
> We sincerely thank you once again for your valuable suggestions and questions. We will update the appendix with additional experiments and further qualitative results.
>
>
> [1] Chen, Youyu, et al. "DashGaussian: Optimizing 3D Gaussian Splatting in 200 Seconds." Proceedings of the Computer Vision and Pattern Recognition Conference. 2025.
>
> [2] Barron, Jonathan T., et al. "Mip-nerf 360: Unbounded anti-aliased neural radiance fields." Proceedings of the IEEE/CVF conference on computer vision and pattern recognition. 2022.
>
> [3] Knapitsch, Arno, et al. "Tanks and temples: Benchmarking large-scale scene reconstruction." ACM Transactions on Graphics.
>
> [4] Kerbl, Bernhard, et al. "3D Gaussian splatting for real-time radiance field rendering." ACM Trans. Graph.
>
> [5] Liu, Yang, et al. "Citygaussian: Real-time high-quality large-scale scene rendering with gaussians." European Conference on Computer Vision.2024.
>
> [6] Wu, Yongchang, et al. "BlockGaussian: Efficient Large-Scale Scene Novel View Synthesis via Adaptive Block-Based Gaussian Splatting." arXiv:2504.09048 (2025).
>
> [7] Kerbl, Bernhard, et al. "A hierarchical 3d gaussian representation for real-time rendering of very large datasets." ACM Transactions on Graphics. 2024.

---

### Official Review · Reviewer_9Q3V · 2025-10-30

**Soundness:** 2
**Presentation:** 2
**Contribution:** 2
**Rating:** 6
**Confidence:** 4

**Summary:**

The paper proposes Signal Structure-Aware Gaussian Splatting (SIG), a framework designed to improve large-scale 3D scene reconstruction using 3D Gaussian Splatting. The authors observe that prior methods supervise Gaussians initialized from sparse low-frequency points using high-frequency image signals, which leads to uncontrolled densification and redundant primitives. To address this, they reformulate scene reconstruction as a signal structure recovery problem and introduce a frequency-based synchronization between image supervision and the Gaussian representation.

Concretely, the paper defines the "average sampling frequency" of images and the "effective scene bandwidth" of Gaussians, derived from a frequency-domain analysis of the opacity field. These metrics guide the adaptive adjustment of training image resolution and Gaussian densification as the scene frequency converges. Additionally, the authors introduce Sphere-Constrained Gaussians (SCG) that restrict each Gaussian’s optimization region to a local sphere determined by its initial point cloud neighborhood, mitigating floaters and structural drift. Experiments on large-scale benchmarks show performance gains in reconstruction quality and training efficiency compared to baselines.

**Strengths:**

The idea of aligning the Gaussian frequency with the supervision signal frequency is intuitive and well-executed, beginning with a clear definition of Gaussian frequency and followed by thorough verification. Experiments show consistent gains in quality and speed, and the method works as a plug-in for existing frameworks like CityGS and BlockGS.

**Weaknesses:**

1. It remains unclear how sensitive the method is to the introduced hyperparameters.
2. While the concept of progressive training from low to high resolution is not novel, the paper’s analytical framework for optimally aligning Gaussian frequency with signal frequency is interesting. However, a comparison with the more heuristic use of progressive training is missing.
3. Evaluating the proposed method on conventional NVS benchmarks such as MipNeRF360 would further demonstrate its generalization capability.

**Questions:**

See the weakness. The LPIPS and SSIM columns appear to be swapped in Tables 2 and 3.

---

> ### Author Response · Authors · 2025-11-21
> **Response to Reviewer 9Q3V - Stage 1**
>
> Thank you for your careful review and valuable comments, which help make our method more convincing and complete. We will address your concerns in detail below.
>
> **W1: Hyperparameter Sensitivity.**
> We agree that hyperparameter selection is crucial for the practical applicability of the method. Below, we provide a detailed explanation and analysis of the rationale behind these choices and their sensitivity.
>
> (1)**Convergence threshold k**: We increase the supervision resolution when
> $ \frac{df}{d\text{iter}} < k $.
> Since different scenes may have inconsistent scales (because COLMAP initialization is not necessarily aligned with real-world scale), we normalize the threshold using
> $ \mathrm{mean}(1/d) $.
> This normalization makes the scales across scenes consistent, ensuring that the threshold $ k $ is independent of the scene scale. We use the same convergence parameter across all large-scale datasets, as
> $ k = 5 \times 10^{-5} $
> works consistently for all five large-scale scenes in the three datasets.
>
> As for **Sensitivity and Selection Principles**, our choice of $k$ follows the principle that it should ensure convergence to the highest frequency before training ends. We observe that the iteration at which the frequency converges varies significantly across datasets. For example, when training at 1/5 resolution on "Rubble", the first Frequency-Aligned state is reached after around 4200 iterations, while in Sci-Art this occurs at about 7800 iterations. This variation arises from the uncertain relationship between the initialized point cloud and the supervisory sampling frequencies. However, as training progresses, the discrepancy between the supervision signal and the reconstructed signal gradually diminishes. We find that all scenes eventually converge to the highest resolution at around 20k iterations.
>
> We also experimented on Rubble with $k = 7 \times 10^{-5}$ and $k = 3 \times 10^{-5}$, and illustrate how different thresholds influence the resolution adjustment. Overall, within a certain range, these variations have little effect on the final evaluation results. (We limit the computation of the frequency to once every 100 iterations, and it is updated only after the frequency change rate stabilizes below $100 \times k$). The table below shows the iteration numbers for different resolution transitions and the final evaluation metrics:
>
> |($\times 10^{-5}$)|1/5 | 2/5  | 3/5  | 4/5   | 5/5   | PSNR  | SSIM  | LPIPS |
> |-----|------|------|-------|-------|-------|-------|-------|-------|
> | 3   | 0    | 2700 | 8100  | 16200 | 18500 | 27.22 | 0.844 | 0.188 |
> | 7   | 0    | 7400 | 9000  | 17500 | 23000 | 27.31 | 0.839 | 0.192 |
> | 5   | 0    | 4200 | 8900  | 17000 | 21000 | 27.35 | 0.843 | 0.189 |
>
>
> *(1/5–5/5 denotes resolutions relative to the final output size. The columns labeled 1/5–5/5 indicate the iteration numbers at which the training transitions to the corresponding resolution.)*
>
> (2) The second key set of hyperparameters is in the **Structure-Aware Optimization**. We explicitly define the optimization bounds for each Gaussian to reduce the occurrence of floaters: $l$ times the average distance of the $K$ nearest neighbors for each Gaussian, i.e., $l \cdot \text{NN}(K)$, where NN(K) denotes the K-nearest neighbor search. Here, NN(K) reflects the density of Gaussians within a local patch around each point. Since $K$ and $l$ have similar effects, $K$ only needs to be consistent across different regions and datasets, whereas $l$ serves as the tunable parameter. Our choice of $l = 15$ is based on the principle that the number of Gaussians pruned in the early stage should generally be less than the number of Gaussians added through densification, ensuring that the scene frequency can increase steadily. We set $l$ to a relatively large value to filter out significantly deviating Gaussians. The results show that $l = 15$ works well across five scenes in three datasets. During the early stages of optimization, when the learning rate is high, these geometric constraints play a crucial role. For example, in Rubble, with an initial count of 1,694,315 Gaussians, the pruning step in the splitting stage removes several hundred to several thousand Gaussians in the early phase. Later in the optimization, this number drops to the order of tens, and regardless of the value of $l$, the final number of pruned Gaussians stabilizes to the order of tens as optimization progresses. We tried different combinations on "Rubble", and the results are shown below.
>
> | l  | K  | final points | PSNR  | SSIM  | LPIPS |
> |----|----|--------------|-------|-------|-------|
> | 5  | 15 | 1.4M         | 27.30 | 0.841 | 0.190 |
> | 15 | 15 | 1.5M         | 27.35 | 0.843 | 0.189 |
> | 15 | 5  | 1.4M         | 27.29 | 0.843 | 0.192 |

---

> ### Author Response · Authors · 2025-11-21
> **Response to Reviewer 9Q3V - Stage 2**
>
> (Continuing from W1)
>
> Additionally, during Gaussian densification, the maximum offset is scaled down to $0.7\times$ (less than 1). The rationale is that densification aims to reconstruct higher-frequency signals, so the optimization space should be further constrained to avoid generating floaters. In our experiments, we found that this parameter has little effect within a certain range, and thus we fix it at $0.7\times$ across all datasets and scenes.
>
> In summary, our method does not require dataset-specific tuning, and all hyperparameters can be directly applied across different datasets. It is robust to varying levels of SfM sparsity. This is partly due to normalization using $\mathrm{mean}(1/d)$ and $\text{NN}(K)$, and partly because varying SfM sparsity mainly affects pruning and scheduling during the early stages of optimization, but these effects stabilize over time and, within a reasonable range, do not significantly affect the final results.
>
> **W2: Comparison with the linear progressive training.**
>
> We sincerely appreciate your valuable suggestions, which substantially improved the completeness of our work. Based on your feedback, we constructed the following baseline. We use the same block partitioning and coarse initialization as our method. During block-wise training, the image resolution is divided into five levels (1/5, 2/5, 3/5, 4/5, and full), and these levels are linearly mapped to the training iterations (0–6000, 6001–12,000, etc.). For densification, we evaluate two strategies. The first follows CityGS[1], performing densification only in the 500-15,000 iterations. The second performs 3,000 densification iterations at each resolution level. On the scene "Rubble", the first strategy yields poor results because Gaussians split only at low resolutions and stop splitting once full resolution is reached, causing a mismatch between the densification and resolution schedulers.The second strategy still suffers from misalignment because it fails to account for the scene’s frequency characteristics, and its final performance is also inferior to the vanilla CityGS. In addition, we include DashGS[2] as a baseline in our main text (**the first row** of Table 1), which also employs a scheduler that linearly maps different image frequencies in the 2D color space to the training iterations. The results are as follows:
>
> | Method      | PSNR  | SSIM  | LPIPS | Points |
> |------------|-------|-------|-------|--------|
> | Strategy-1 | 26.21 | 0.804 | 0.240 | 1.4M   |
> | Strategy-2 | 26.39 | 0.811 | 0.240 | 1.6M   |
> | DashGS     | 26.37 | 0.802 | 0.237 | 1.3M   |
> | Ours       | 27.35 | 0.843 | 0.189 | 1.5M   |
>
> **W3: Generalization evaluation on other standard datasets.**
>
> We sincerely appreciate your insightful suggestion. A robust method should indeed generalize across different environment settings. Although our scheduler was originally designed for large-scale scene reconstruction, our approach remains effective on small-scale scenes as well. Following your recommendation, we conducted additional experiments on the Mip-NeRF360[3] and Tanks-and-Temples[4] datasets.
>
> For small-scale scenes, the divide-and-conquer strategy used in large-scale scenarios is unnecessary. Because our method is fully plug-and-play, we use the original vanilla 3DGS[5] as the baseline and report 3DGS + ours as our method, and all training configurations strictly follow the original 3DGS settings without any modification. In addition, our hyperparameters remain identical to those used for large-scale scenes.
>
> Surprisingly, although our method was initially designed for large-scale reconstruction, it still significantly improves both the performance and efficiency of vanilla 3DGS on small-scale datasets. Our approach achieves nearly 2× training speedup, and outperforms methods of DashGS and vanilla 3DGS. On the Tanks-and-Temples dataset, we evaluate on the Train and Truck scenes; on the Mip-NeRF360 dataset, we evaluate on the Flowers and Bicycle scenes. The results are as follows (**Qualitative results are provided in Appendix 6 in the updated version**):
>
>
>
> |**Tanks-and-Temples**         | Train |      |       | Truck |      |       |
> |---------|-------|------|-------|-------|------|-------|
> | Method  | PSNR  | SSIM | LPIPS | PSNR  | SSIM | LPIPS |
> | 3DGS    | 21.09 | 0.80 | 0.22  | 25.19 | 0.88 | 0.15  |
> | DashGS  | 22.01 | 0.80 | 0.22  | 25.66 | 0.88 | 0.16  |
> | Ours    | 22.33 | 0.82 | 0.20  | 25.90 | 0.88 | 0.15  |
>
> | **Mip-NeRF360**        | Flowers |      |       | Bicycle |      |       |
> |---------|-------|------|-------|-------|------|-------|
> | Method  | PSNR  | SSIM | LPIPS | PSNR  | SSIM | LPIPS |
> | 3DGS    | 21.52 | 0.61 | 0.34  | 25.25  | 0.77 | 0.21  |
> | DashGS  | 22.17 | 0.63 | 0.32  | 25.70  | 0.78 | 0.20  |
> | Ours    | 22.20 | 0.64 | 0.33  | 25.77  | 0.78 | 0.21  |

---

> ### Author Response · Authors · 2025-11-21
> **Response to Reviewer 9Q3V - Stage 3**
>
> (Continuing from W3)
>
> Although our method was originally motivated by the significant mismatch between the frequencies of the sparsely initialized Gaussians and the sampling frequency in large-scale scenes, this phenomenon also persists in small-scale scenes. This occurs because, while the SFM-initialized point cloud in small-scale scenes indeed exhibits higher initial frequencies, the substantially higher sampling density leads to a correspondingly larger sampling frequency.
> For example, in the truck scene from Tanks and Temples, the average Gaussian frequency increases from 9.07 at initialization to 35.92 after convergence (compared with 3.23 to 14.35 in the large-scale scene rubble). Therefore, the frequency-aligned strategy remains effective even in small-scale scenarios.
>
> Additionally, we noticed that the order of SSIM and LPIPS was incorrect in Tables 2 and 3, and this error has been corrected in the latest version. We would like to sincerely thank you once again for your valuable suggestions and questions. The additional experimental results and further qualitative results are provided in the appendix.
>
> [1] Liu, Yang, et al. "Citygaussian: Real-time high-quality large-scale scene rendering with gaussians." European Conference on Computer Vision. 2024.
>
> [2] Chen, Youyu, et al. "DashGaussian: Optimizing 3D Gaussian Splatting in 200 Seconds." Proceedings of the Computer Vision and Pattern Recognition Conference. 2025.
>
> [3] Barron, Jonathan T., et al. "Mip-nerf 360: Unbounded anti-aliased neural radiance fields." Proceedings of the IEEE/CVF conference on computer vision and pattern recognition. 2022.
>
> [4] Knapitsch, Arno, et al. "Tanks and temples: Benchmarking large-scale scene reconstruction." ACM Transactions on Graphics (ToG) 36.4 (2017): 1-13.
>
> [5] Kerbl, Bernhard, et al. "3D Gaussian splatting for real-time radiance field rendering." ACM Trans. Graph. 42.4 (2023): 139-1.

---

### Author Response · Authors · 2025-11-30
**Summary Note to the Area Chair**

We sincerely appreciate the reviewers’ time and thoughtful feedback on our submission. Below, we provide a global summary. We begin by outlining the main contributions of our work. We then summarize both the reviewers’ positive assessments and their primary concerns, followed by an overview of our responses.

**(1) Our contributions:**

We rethink the scheduler for 3DGS training from the perspective of signal structure recovery. Based on this viewpoint:
- We mathematically define the average frequency of the 3D Gaussian representation and introduce a novel scheduler that synchronizes image supervision with Gaussian frequency, effectively reducing redundancy and accelerating training.
- We propose Sphere-Constrained Gaussians, which leverage geometric priors to restrict the optimization space.
- Our framework delivers substantial improvements in both reconstruction quality (+0.9 dB PSNR) and training efficiency (1.5× per block) across multiple benchmarks.

**(2) Summary of Positive Feedback from Reviewers:**
- Reviewers 9Q3V, bS7H, and aMuM generally found the idea of aligning the Gaussian frequency with the supervision signal frequency to be intuitive.
- All reviewers acknowledged that our method shows promising results in improving the efficiency and quality of large-scale scene reconstruction.
- Furthermore, the potential plug-and-play utility of our approach was noted by 9Q3V and P2B2. Reviewers P2B2, bS7H, and aMuM also commented positively on the clarity of our writing and experimental presentation.

**(3) Summary of the reviewers’ key suggestions and concerns:**
- Hyperparameter Sensitivity and Selection. (9Q3V, P2B2)

Following the suggestions of reviewers 9Q3V and P2B2, we have addressed their concerns from two aspects:

1. Providing a comprehensive analysis of the rationale behind our hyperparameter choices.

2. Conducting a systematic evaluation of Hyperparameter sensitivity, including the effects of their variations on performance.

Importantly, our method does not require dataset-specific tuning; all hyperparameters can be directly applied across different datasets. **We conducted experiments to assess the sensitivity of all key hyperparameters and reported the results**. The method demonstrates robustness to different datasets and even different types of scenes. This robustness is partly due to the normalization using $\mathrm{mean}(1/d)$ and $\text{NN}(K)$ in Eq.6 of main text, and partly because variations in SfM sparsity primarily influence pruning and scheduling during the early stages of optimization. These effects stabilize over time and, within a reasonable range, do not significantly affect the final results.

- Comparison with the linear progressive training (9Q3V, bS7H)

Following the recommendations of reviewers 9Q3V and bS7H, we implemented linear progressive training combined with two distinct densification strategies. Additionally, the hard-coded baseline DashGS, as compared in the main text, is also included below. The results further demonstrate the effectiveness of our method.

| Method      | PSNR  | SSIM  | LPIPS | Points |
|------------|-------|-------|-------|--------|
| Strategy-1 | 26.21 | 0.804 | 0.240 | 1.4M   |
| Strategy-2 | 26.39 | 0.811 | 0.240 | 1.6M   |
| DashGS     | 26.37 | 0.802 | 0.237 | 1.3M   |
| Ours       | 27.35 | 0.843 | 0.189 | 1.5M   |

- Generalization evaluation on other standard datasets (9Q3V, bS7H)

Following the recommendations of reviewers 9Q3V and bS7H, we conducted additional experiments on the Mip-NeRF360 and Tanks-and-Temples datasets.

|**Tanks-and-Temples**         | Train |      |       | Truck |      |       |
|---------|-------|------|-------|-------|------|-------|
| Method  | PSNR  | SSIM | LPIPS | PSNR  | SSIM | LPIPS |
| 3DGS    | 21.09 | 0.80 | 0.22  | 25.19 | 0.88 | 0.15  |
| DashGS  | 22.01 | 0.80 | 0.22  | 25.66 | 0.88 | 0.16  |
| Ours    | 22.33 | 0.82 | 0.20  | 25.90 | 0.88 | 0.15  |

| **Mip-NeRF360**        | Flowers |      |       | Bicycle |      |       |
|---------|-------|------|-------|-------|------|-------|
| Method  | PSNR  | SSIM | LPIPS | PSNR  | SSIM | LPIPS |
| 3DGS    | 21.52 | 0.61 | 0.34  | 25.25  | 0.77 | 0.21  |
| DashGS  | 22.17 | 0.63 | 0.32  | 25.70  | 0.78 | 0.20  |
| Ours    | 22.20 | 0.64 | 0.33  | 25.77  | 0.78 | 0.21  |

Although motivated by the frequency mismatch in large-scale scenes, our method still significantly improves both the performance and efficiency of vanilla 3DGS on small-scale datasets.This occurs because the higher sampling density in small-scale scenes results in a higher sampling frequency that still significantly exceeds the initial Gaussian frequencies. For example, in the truck scene from Tanks and Temples, the average Gaussian frequency increases from 9.07 at initialization to 35.92 after convergence (compared with 3.23 to 14.35 in the large-scale scene rubble). Therefore, the frequency-aligned strategy remains effective even in small-scale scenarios.

---

### Meta-Review · Area_Chair_S9Ga · 2026-01-07

**Summary:**

This paper is a clear accept from the initial reviews. It receives 3x marginal accepts and 1x accept. The strengths are 1) the idea is novel; 2) the writing is good; 3) the experiments clearly shows the performances and justified the design choices. The weaknesses raised by the reviewers are minor. They asked mostly for further clarifications of experiments and some further comparisons to strengthen the paper. The AC follows the suggestions of the reviewers to accept the paper.

**Reviewer Concerns:**

The weaknesses raised by the reviewers are minor. They asked mostly for further clarifications of experiments and some further comparisons to strengthen the paper. These concerns are well-addressed in the rebuttal.

**Reviewer Scores:**

The reviewers will still maintain their positive scores.

---

### Decision · Program_Chairs · 2026-01-26

Accept (Poster)